# Minimally Invasive Facetectomy and Fusion for Resection of Extensive Dumbbell Tumors in the Lumbar Spine

**DOI:** 10.3390/medicina58111613

**Published:** 2022-11-08

**Authors:** Michael Schwake, Emanuele Maragno, Marco Gallus, Stephanie Schipmann, Dorothee Spille, Bilal Al Barim, Walter Stummer, Michael Müther

**Affiliations:** 1Department of Neurosurgery, University Hospital Münster, Albert-Schweitzer-Campus 1, Gebäude A1, 28149 Münster, Germany; 2Department of Neurosurgery, Haukeland University Hospital, 5021 Bergen, Norway

**Keywords:** dumbbell tumors, minimally invasive surgery, facetectomy, nerve sheet tumors, schwannoma

## Abstract

*Background and Objectives*: Resection of dumbbell tumors can be challenging, and facet joint sparing approaches carry the risk of incomplete resection. In contrast, additional facetectomy may allow better surgical exposure at the cost of spinal stability. The aim of this study is to compare facet-sparing and facetectomy approaches for the treatment of lumbar spine dumbbell tumors. *Materials and Methods*: In a cohort study setting, we analyzed Eden type 2 and 3 tumors operated in our department. Conventional facet-sparing microsurgical or facetectomy approaches with minimally invasive fusions were performed according to individual surgeons’ preference. Primary outcomes were extent of resection and tumor progression over time. Secondary outcomes were perioperative adverse events. *Results*: Nineteen patients were included. Nine patients were operated on using a facet-sparing technique. Ten patients underwent facetectomy and fusion. While only one patient (11%) in the facet-sparing group experienced gross total resection (GTR), this was achieved for all patients in the facetectomy group (100%). The relative risk (RR) for incomplete resection in the facet-sparing cohort was 18.7 (95% CI 1.23–284.047; *p* = 0.035). In addition, time to progression was shorter in the facet-sparing cohort (*p* = 0.022) and all patients with a residual tumor underwent a second resection after a median follow-up time of 42 months (IQR 25–66). *Conclusions*: Minimally invasive resection of lumbar Eden type 2 and 3 dumbbell tumors including facetectomy in combination with instrumentation appears to be safe and superior to the facet-sparing approach in terms of local tumor control.

## 1. Introduction

Dumbbell tumors are usually benign and extent though the neuroforamen. Symptomatic tumor compression of neural structures and tumor progression are frequent indications for resection. The architecture of these tumors, especially the Eden types 2 and 3 [1,2] in the lumbar spine, involving both the spinal canal and paraspinal structures make tumor resection challenging, particularly when executing minimally invasive approaches [2,3].

In recent years, several techniques for the resection of dumbbell tumors have been introduced. Importantly, it remains controversial whether facetectomy is necessary to achieve gross total resection (GTR). Facet-sparing techniques may harbor the risk of incomplete tumor resection due to limited exposure of the neuroforamen. Yet, a facetectomy in the lumbar spine can cause spinal instability and necessity for additional fusion. In addition to various case series, the literature is lacking comparative studies on higher levels of evidence [2,3,4,5].

The aim of this study is to compare two common surgical procedures. The resection via unilateral facet-sparing hemilaminectomy is considered the microsurgical mainstay. In contrast, a more advanced technique adds facetectomy for wider exposure and subsequent minimally invasive instrumentation. Primary outcomes were extent of resection and tumor progression over time. Secondary outcomes were perioperative adverse events.

## 2. Materials and Methods

### 2.1. Study Design and Setting

This is an observational cohort study comparing two techniques of spinal dumbbell tumor resection in a single academic center setting. Since 2018, data were collected prospectively as part of the registry of the German Spine Society (DWG). Reporting is in accordance with Strengthening the Reporting of Observational Studies in Epidemiology (STROBE) guidelines for cohort studies [6]. The study was conducted in accordance with the Declaration of Helsinki. Ethical approval was obtained from the institutional review board (case numbers 2020e620-f-S and 2016-045-b-S). The setting is a single center. 

### 2.2. Participants

All spinal dumbbell tumors, Eden type 2 and 3, operated at the neurosurgical department, University Hospital Münster between 2014 and 2021, were included [1,2]. The groups of facet-sparing and facetectomy approaches were formed according to individual surgeons’ preferences. See Figure 1 for a study flow diagram. 

### 2.3. Data and Variables

The electronic hospital information system was searched for the following data points: age, sex, symptoms, disease status (de-novo tumor versus progressive disease), hospital length of stay (LOS), and tumor growth according to Eden classification: type 1 (intra- and extradural), type 2 (intra- and extradural) and paravertebral, type 3 (extradural and paravertebral), and type 4 (foraminal and paravertebral) [1,2]. Only patients with Eden types 2 and 3 were included in this study. Preoperative semiautomatic tumor volumetric analysis via segmentation was performed using Brainlab elements^®^ software (Brainlab AG, Munich, Germany) and expressed in milliliters (mL). Additionally, we documented perioperative adverse events and intraoperative blood loss (mL). Follow-up magnet resonance imaging (MRI) was scheduled three months after surgery in all cases as baseline scan. This was followed by annual MRI for benign pathologies. Extent of resection was classified as either gross total resection (GTR, no residual contrast enhancing tumor on postoperative MRI) or subtotal resection (STR). Follow-up MRI focused on the detection of tumor recurrence and indirect signs of non-fusion-like screw loosening, hardware failure or pain. 

Functional outcome was assessed according to the Macnab criteria 6 months after surgery [7]. For further analysis, we categorized Macnab score 4–5 as favorable and 1–3 as unfavorable functional outcome. MRI scans and clinical functionality were determined by two blinded investigators (MS, MM).

### 2.4. Surgical Techniques

Patients in the facetectomy cohort were operated as described previously [8]. Briefly, carbon fiber-reinforced polyether ether ketone (CFR-PEEK; VADER; icotec AG, Altstätten, Switzerland) pedicle screws were inserted percutaneously. Afterwards, on the side of the tumor, the two pedicle screw incisions were joined. Subsequently, a transmuscular retraction device (Terra Nova; Stryker, Kalamazoo, MI, USA) was connected to the screws to prepare the surgical field for posterolateral facetectomy and laminotomy, followed by microsurgical tumor resection (see Figure 2). In Eden type 2 lesions, the intradural part of the tumor was resected via durotomy, which was closed subsequently with a continuous suture and TachoSil^®^ (Takeda, Tokyo, Japan) sealing. Discectomy and cage (icotec AG) insertion via trans-foraminal approach was performed if the index level was below L 2/3. Autologous bone and artificial bone adjuvant (Cerasorb foam; Curasan AG, Kleinostheim, Germany) were applied unilaterally to bridge the facetectomy gap. Finally, CFR-PEEK rods (BlackArmor; icotec AG) were inserted and fixed to the pedicle screws. In this group, early postoperative 1.5-T magnetic resonance imaging (MRI) was performed within 48 h after surgery to check implant placement and to serve as a basis for further imaging.

In the facet-sparing cohort, tumor resection was performed via ipsilateral hemilaminectomy via a Caspar retraction device (B. Braun, Tuttlingen, Germany). The paraspinal part of the tumor was resected via the dorso-lateral approach, if needed, also using the Caspar retractor. Again, in case of Eden type 2 lesion, the intradural part of the tumor was resected via durotomy as described above. In all cases, nerve roots were identified using direct nerve stimulation, and spared whenever possible.

All procedures were performed under intraoperative neurophysiological monitoring (IOM), including somatosensory (SSEP) and motor evoked potentials (MEP), as well as continuous electromyography (EMG) and direct nerve stimulation. 

### 2.5. Statistics

Statistical analyses were performed using the software IBM SPSS Statistics 24.0 (IBM Corp., Armonk, NY, USA). Categorical variables are shown as absolute and relative frequencies. The Schapiro-Wilk test was performed to evaluate whether variables are normally distributed. Parametric values are presented in mean and standard deviation (SD). Non-parametric values are presented as median and interquartile range (IQR, 25% quartile and 75% quartile). Two-tailed Student t-tests was used as parametric and two-sided Mann-Whitney U-tests (MWU) as non-parametric test. Fisher’s exact test was performed to compare groups of categorical variables. Kaplan–Meier curve was used to plot time to progression and curves were compared using log rank test. In addition, relative risk (RR) of incomplete resection (subtotal resection), including 95% confidence intervals (CI), was calculated for the facet-sparing cohort. A probability value less than 0.05 was considered statistically noticeable throughout the whole analyses.

## 3. Results

From 2014 to 2021, 82 intra-spinal and extra-medullary lumbar tumors were treated surgically at our department. In nineteen (8%) cases, the tumor had an Eden type 2 or 3 dumbbell configuration. In eleven cases—since 2018—the data were collected prospectively (see Appendix A). Each group included one WHO grade II hemangiopericytoma; all other tumors were WHO grade I schwannomas (Table 1). 

Resection was performed according to the surgeon’s preference. In nine cases, resection was conducted via unilateral hemilaminectomy, including a dorsolateral approach and thereby sparing the facet joint in order not to cause spinal instability. In the other 10 cases, tumor resection was carried out via a minimally invasive facetectomy, partial hemilaminectomy und percutaneous fusion, as described previously [8]. The two cohorts did not show any statistically noticeable differences in patient characteristics (Table 1). 

Gross total resection (GTR) of the tumor was achieved in all cases within the facetectomy cohort (N = 10, 100%) in comparison to only one case in the facet-sparing cohort (N = 1, 11.1%), The relative risk of incomplete resection in the facet-sparing cohort was 18.7 (CI 95% [1.23, 284.047], *p* = 0.035). Duration of surgical procedure, intraprocedural blood loss, and length of hospital stay (LOS) were distributed equally in both groups. One case of unexplained transient monoparesis to the leg was noted in the facet-sparing cohort. Otherwise, we did not record any relevant postoperative adverse event (Table 2).

During further median follow-up time (51 months IQR: 21–75), all patients with residual tumor (N = 8, 88.9%, *p* < 0.001) showed radiographic signs of tumor progression and underwent a second resection after a median follow up time of 42 months (IQR 25–66). In comparison, all patients with GTR—in both groups—are still tumor free after a median follow up time of 24 months (IQR 14–40) (see Table 2). Kaplan–Meier statistics revealed a significantly shorter time to progression for the facets-sparing cohort χ²(1) = 5.273, *p* = 0.022; Figure 3).

When analyzing the functional outcome post-operatively, we noticed that in the facetectomy cohort, four patients (40%) had an excellent outcome (Macnab 5), while six (60%) had a good outcome (Macnab 4). In the facet-sparing group, most patients (N = 7, 77.7%) also had a good outcome, three (33.3%) of those were Macnab 4 (excellent). However, two (22.2%) patients were scored unfavorable Macnab 2 and 3. Fisher exact test did not reveal a significant difference in functional outcome between the cohorts when grouping for favorable (Macnab 4–5) and unfavorable (Macnab 1–3) (*p* = 0.19) (Table 2), Figure 4 and Figure 5:

In the further follow-up period, (median: 21 months, IQR 14–37), we did not record any implant-related complications such as misplacement, migration, pseudarthrosis, adjacent level disease (ALD) or proximal junctional kyphosis (PJK) within the facetectomy cohort.

## 4. Discussion

The minimally invasive approach including facetectomy and percutaneous instrumentation allowed for a higher rate of complete tumor resection of Eden type 2 and 3 dumbbell tumors in the lumbar spine, which was achieved in all cases (N = 10, 100%, *p* < 0.001). This translated into a relative risk of 18.70 (CI 95% [1.23, 284.047], *p* = 0.035) for incomplete resection in the facet-sparing cohort. No patient in the facetectomy group required a second surgery due to tumor progression (*p* = 0.022). Furthermore, facetectomy did not result in a higher rate of procedure-related early perioperative or late implant-related complications. Our results highlight the importance of unlocking deep compartments such as the neuroforamen to achieve favorable oncological and functional outcomes in Eden type 2 and 3 lesions, even at a cost of instability, which was subsequently addressed by minimally invasive instrumentation. 

Eden type 2 and 3 lesions form the largest groups of spinal dumbbell tumors [2]. Resection of these tumors can be surgically challenging with high rates of subtotal resection [4]. Surgical methods, such as laminectomy, hemilaminectomy, or lateral transforaminal approaches carry the risk of incomplete resection. Therefore, some authors recommend additional facetectomy to achieve complete resection [2,9]. We believe that the main rationale for facetectomy is optimal visualization of the tumor, its blood supply and identification of the nerve root in the neuroforamen. However, facetectomy especially in the lumbar spine may cause spinal instability and additional fusion is advocated by several authors [2,10]. Furthermore, open surgeries including laminectomy, facetectomy, tumor resection and instrumentation may lead to significant muscle trauma, higher blood loss, and a higher rate of wound infections and pain [11,12], which may lead to a longer period of recovery. 

With this study, we demonstrate that our minimally invasive facetectomy and percutaneous instrumentation did not result in significantly longer operating times when compared to a standard microsurgical approach. Moreover, we did not notice higher blood loss or longer length of hospital stay. Finally, we noticed neither surgery-related complication such as new neurological deficits, surgical site infections nor leakage of cerebrospinal fluid. Functional outcome in the facetectomy group, according to the Macnab score, did not differ significantly compared to the facet-sparing group. This shows that additional fusion after tumor resection did not cause any relevant impairment to patients’ functional outcome. All patients in the facetectomy group returned to full employment after surgery. 

Previous case series underline the eminent role of GTR to prevent progressive disease and to maintain good long-term outcome. Many studies further showed that STR, even of benign tumors such as schwannomas, may carry the risk of tumor progression [5,9,13,14,15,16]. This emphasizes the necessity of achieving GTR whenever possible, also in the prize of facetectomy, potential instability and more invasive surgery. 

Nevertheless, it remains controversial whether facetectomy is necessary in order to achieve GTR of these tumors. Indeed, recent studies demonstrated excellent results in the case of Eden type 3 and 4 tumors involving the paraspinal space and neuroforamen alone via minimally invasive dorsolateral and endoscopic approaches without facetectomy [17,18]. However, both studies had a short follow up and included neither large volume tumors, those involving the vertebral bodies, the intra-dural space, nor recurrent tumors or ones with abundant blood supply. In an additional case series, Poblete et al. [19] found a combined interlaminar and far lateral facet sparing approach sufficient for complete resection of Eden type 3 tumors. However, their study does not include intradural, type 2, tumors. Moreover, their series had only seven, all primary cases, in the lumbar spine, which were relatively smaller showing no expansion beyond one vertebral level. Moreover, in comparison with the above-mentioned studies [17,18,19], we have longer follow-up times, especially in the non-facetectomy cohort, showing tumor progression requiring second surgery after a mean time of 42 months. To the best of our knowledge, our study is the first one comparing both surgical approaches in a cohort setting.

Concordantly to our hypothesis, Poblete et al. assume that that main issue of concern is the part of the tumor inside the neuroforamen [19]. However, in contrast, we believe that visual and manual inspection without facetectomy is not always sufficient and facetectomy is required to confirm complete resection of the tumors. One further important issue is that postoperative images are not easy to interpret, because of the contrast enhancement of scar tissue in the neuroforamen after tumor resection, making identification of small tumor remnants difficult. This makes the role of direct visual inspection of these remnants very important. 

Another important issue in the management of intraspinal tumors is postoperative surveillance. Because of the potential risk of tumor progression, magnet resonance imaging (MRI) should be performed on a regular basis. However, spinal instrumentation with standard titanium implants may cause significant artefacts on MRI, which makes interpretation very difficult. To prevent this, we utilized CFR-PEEK hardware. This minimally metallic instrumentation (MMI) causes less artifacts, thus allowing for more adequate detection of potential tumor regrowth or other perioperative morbidities such as hematoma or infection [8]. Moreover, MMI may also be very helpful in case adjuvant irradiation is required. Previous studies have already demonstrated the benefits of MMI in the planning and execution of photon as well as particle radiation therapy [20,21,22,23]. Similar mechanical properties and fusion rates compared to titanium implants have been well documented in the literature [24,25,26]. 

A major limitation of our study is the relatively small size of cohorts and lack of randomization. Still, dumbbell tumors in the lumbar spine form a small group of only approximately 20% [2,11] of all dumbbell tumors, which again represent approximately 20% of all intra-spinal tumors. Yet, to our knowledge this work forms one of the largest studies concerning this pathology. We were not able to provide data on quality of life or patient reported outcome measures. We will be continuing to enhance our registry with new cases which will hopefully allow for the analysis of larger patient numbers in the future.

## 5. Conclusions

Resection of lumbar dumbbell tumors including facetectomy in combination with percutaneous instrumentation with CFR-PEEK implants appears to be safe and superior to the standard microsurgical facet-sparing approach in terms of local tumor control and functional outcome. Analyzing larger patent cohorts will be necessary to confirm these results.

## Figures and Tables

**Figure 1 medicina-58-01613-f001:**
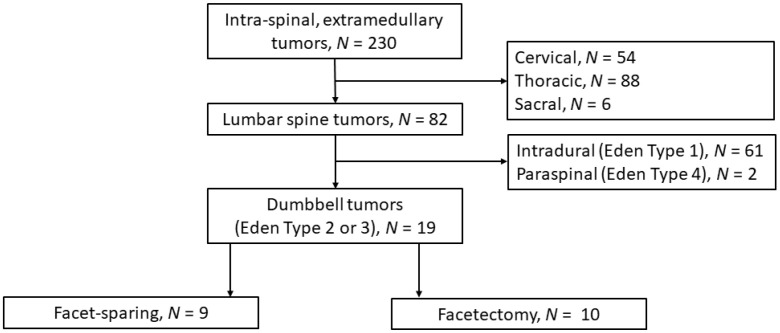
Flow chart of patients included to the study.

**Figure 2 medicina-58-01613-f002:**
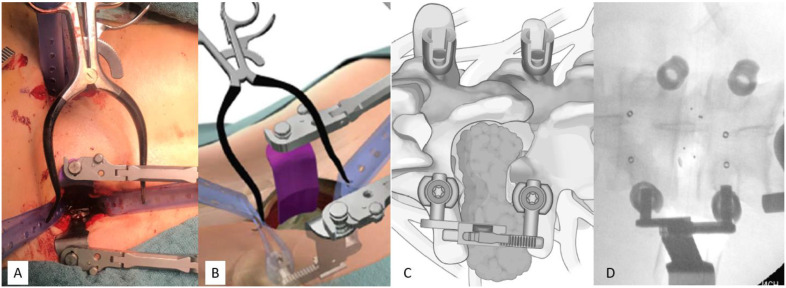
Intraoperative setting of the minimally invasive facetectomy approach. After percutaneous insertion of the pedicle screws, the trans muscular retraction device is connected to the pedicle screws (**A**,**B**). Via this posterolateral approach laminotomy and facetectomy allow for complete visualization and resection of the dumbbell tumor (**C**). (**D**) demonstrates an intraoperative fluoroscopy.

**Figure 3 medicina-58-01613-f003:**
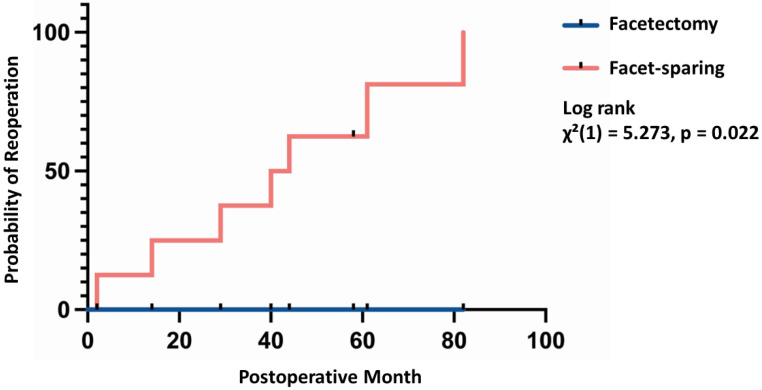
Cumulative risk of reoperation over time for patients with lumbar dumbbell tumors. Blue line indicates patients in the facetectomy group (N = 10, median follow up 21 months (IQR 14–37) and the red line patients in the facet-sparing group (N = 9, median follow up 44 moths (IQR 29–61). Log rank Meier-Kaplan analysis shows a higher risk of reoperation in the facet-sparing group (*p* = 0.022).

**Figure 4 medicina-58-01613-f004:**
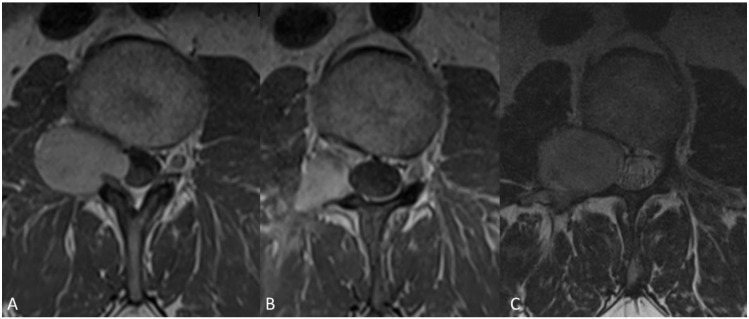
Case example of a 46-year-old male patient admitted with sciatica and low back pain. MRI demonstrated a space occupying lesion at the level of L 2–3, Eden type 3 (**A**), histology conformed a grade I schwannoma. Tumor resection was performed via dorso-lateral approach. Post-operative MRI, three months after surgery showed reduction of tumor volume with residual contrast enhancement (**B**) and patient was followed up. After a period of three years MRI demonstrated clear tumor progression (**C**) and patient had to be operated again.

**Figure 5 medicina-58-01613-f005:**
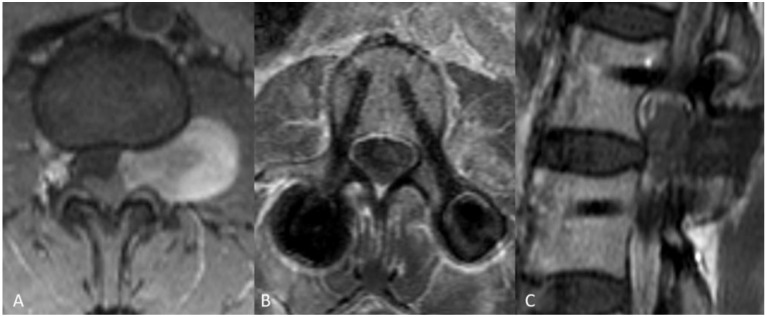
Case example of a 31-year-old female patient with low back pain. MRI showed a contrast enhancing space occupying lesion at L 2–3, Eden type 3 (**A**), tumor resection was performed via facetectomy and minimally invasive instrumentation. Postoperative images (**B**,**C**) show complete tumor resection. Notice the clear images with minimal artefacts due to CFR-PEEK instrumentations. Histology confirmed a grade I schwannoma.

**Table 1 medicina-58-01613-t001:** Baseline patient characteristics.

	Facetectomy, N = 10	Facet-Sparing, N = 9	*p* Value
M/F (N)	6/4	8/1	0.3034
Median age (IQR)	49 years (30–53)	38 years (33–46)	0.596
Median surgery time (IQR)	246 min (218–278)	215 min (139–250)	0.34
Median tumor volume (IQR)	9.64 mL (5.13–15.40)	7.75 mL (4.37–28.90)	0.39
Tumor Eden type 2	2	2	1
Tumor Eden type 3	8	7	1
Primary surgery	4 (40%)	7 (78%)	0.17
Side (left/right)	7/3	6/3	1
Pathology			
Schwannoma (WHO grade I)	9 (9%)	8 (89%)	1
Hemangiopericytoma (WHO grade II)	1 (10%)	1 (11%)	

**Table 2 medicina-58-01613-t002:** Perioperative data and outcome.

	Facetectomy	Facet-Sparing	*p* Value
Median blood loss (IQR)	200 mL (75–875)	100 mL (50–300)	0.466
Gross total resection (N, 10%)	10 (100%)	1 (11.1%)	<0.001
Number of complications	0	1	
Median length of hospital stay (IQR)	4 days (3–6)	6 days (5–8)	0.121
Macnab score 5 (excellent outcome)	4 (40%)	3 (33.3%)	
Macnab score 4 (good outcome)	6 (60%)	4 (44.45%)	
Macnab score 3 (fair outcome)		1 (11.1%)	
Macnab score 2 (poor outcome)		1 (11.1%)	
Favorable outcome (Macnab score 4–5)/ Unfavorable outcome (Macnab score 1–3)	10/0 (100%)	7/2 (77.7%/22.3%)	0.19
Second surgery required (%)	N = 0, 0%	N = 8 (88.9%)	<0.001
Mean follow up time	21 (IQR 14–37)	44 (IQR 29–61)	0.171

## Data Availability

Data can be found at our institution.

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
