# Peer review of "Minimally Invasive Facetectomy and Fusion for Resection of Extensive Dumbbell Tumors in the Lumbar Spine"

_medicina, 2022, doi:10.3390/medicina58111613_

Round 1
Reviewer 1 Report (Previous Reviewer 1)
No further comments
Reviewer 2 Report (New Reviewer)
This is an interesting study showing safety and efficacy of minimally invasive facetectomy and fusion for resection of dumbbell tumors in the lumbar spine, as compared to facet-sparing approach.
Reviewer 3 Report (New Reviewer)
In this paper Schwake et al. report that minimally invasive resection of lumbar Eden type 2 and 3 dumbbell tumors including facetectomy in combination with instrumentation appears to be safe and superior to the facet-sparing approach in terms of local tumor control. Although it does not represent an innovative and original topic the manuscript seems well structured. The methods are relevant to the topic, tabs and figures are exhaustive and discussion is quite detailed.
The topic is quite interesting, especially considering the focus on minimally invasive procedures, as well as the intent for gross total resection during treatment of benign lesions. These two aspects not always run on the same direction.
The only relevant limitation, as confirmed by the Authors, is the lack of randomization that might have had some impact in the final results.
I do not have any specific correction to recommend.
This manuscript is a resubmission of an earlier submission. The following is a list of the peer review reports and author responses from that submission.
Round 1
Reviewer 1 Report
1) Is there any reason why the authors didn't included Eden type IV tumors? If so, this should be clarified in the text, as type IVB schwannomas are considered dumbbell type.
2) Surgical technique should indicate differences between both surgical techniques with nerve root management, if nerve root stimulation was used and if this influenced the surgeon with the decision of making or not a fascetectomy.
3) I suggest to describe results with giant dumbbell schwannomas (>2.5cm of extra spinal extension or more than 2 contiguous vertebral bodies) and if this influenced the decision between both techniques, and if this variable influenced surgical results.
4) Suggest to revise surgical series with fascet sparing techniques, as DOI https:doi.org/10.1055/s-0041-1739502, and compare results.
Author Response
We thank the reviewer for their comments:
In the following, we answer the questions point by point:
- Is there any reason why the authors didn't included Eden type IV tumors? If so, this should be clarified in the text, as type IVB schwannomas are considered dumbbell type.
According to the literature and personal experiences, Type 4 lesions may well be resected via a lateral approach. Hence, we decided to include only Type 2 and 3 tumors. Also, Eden 4 tumors accounted only for two cases in our cohort, which does not allow for a meaningful analysis. In lines 35 -36 we highlighted that resection of Type 2 and 3 tumors is most challenging due to the involvement of both intra- and paraspinal structures.
- Surgical technique should indicate differences between both surgical techniques with nerve root management, if nerve root stimulation was used and if this influenced the surgeon with the decision of making or not a fascetectomy.
We used electrophysiological nerve root monitoring in both settings and nerve roots were spared whenever possible, especially when stimulation suggested motor function.
This was now clarified in the methods section.
- I suggest to describe results with giant dumbbell schwannomas (>2.5cm of extra spinal extension or more than 2 contiguous vertebral bodies) and if this influenced the decision between both techniques, and if this variable influenced surgical results.
- We compared volumetric tumor size between groups and did not find a statistically significant difference. We highlighted that despite tumor size, the most challenging aspect of resecting dumbbell tumors is adressing the intraforaminal part. Suggest to revise surgical series with fascet sparing techniques, as DOI https:doi.org/10.1055/s-0041-1739502, and compare results.
We thank the reviewer for pointing out this publication, which we gratefully included as a citation. We agree that resection of large extra-spinal tumors can be achieved by a lateral approach and that the main issue of concern is the intraforaminal part of the tumor underneath the pars articularis. Still, we feel that the series is not fully comparable to our cohorts as follow up is shorter, intradural tumors were not included and only half of the patients suffered from lumbar lesions.
Reviewer 2 Report
Comments to the Author
I thank you for the opportunity to review your paper.
The authors evaluated the facet-sparing and facetectomy approaches for the treatment of lumbar spine dumbbell tumors. Nineteen patients were included. Nine patients were operated on using a facet-sparing technique. Ten patients underwent facetectomy and fusion. The relative risk (RR) for incomplete resection in the facet-sparing cohort was 18.7 (95% CI 1.23 - 284.047; p=0.035). The results showed that minimal invasive resection of lumbar Eden type 2 and 3 dumbbell tumors, including facetectomy in combination with instrumentation, appears safe and superior to the facet-sparing approach in local tumor control.
This is a topic of interest to spine surgeons. However, this paper has several issues to be resolved, such as sample size and indications in the surgical procedures, as the author mentions in the limitations. In order to turn this data into a valuable paper, it is necessary to increase the number of cases and decide on indications for surgical procedures. For example, what tumor size should a facetectomy approaches be performed for GTR?
Moreover, please present a case, including preoperative and postoperative X-ray and MRI images.
My comments are below.
Line 72-73; “Preoperative tumor volumetric analysis was performed using Brainlab elements® (Brainlab AG, Munich, Germany) and expressed in milliliters (ml).” Please be specific, and it is easy to understand with a diagram. I think the tumor size may have affected the surgical technique.
2.4 Surgical techniques; -There are no indications for the two procedures. How did you make each surgical decision?
The authors describe this in the results as follows. “Resection was performed according to surgeon's preference in line 139” This seems like a scientific problem.
Line 131-132, “From 2014 to 2021 two hundred and thirty (N=230) patients were treated surgically at our department for intraspinal extramedullary tumors. Of these, 82 were in the lumbar spine.” We don't need this data, and it should be removed.
Line 147- 150, “Gross total resection (GTR) of the tumor was achieved in all cases within the facetectomy cohort (N=10, 100%) in comparison to only one case in the facet-sparing cohort (N=1, 148 11,1%).” This result inevitably has a significant impact on the recurrence rate.
In the first place, there is a problem in comparing surgical results between two groups in which complete resection has not been performed.
Line 203-204, “To the best of our knowledge, our study is the first comparative analysis on both 203 surgical approaches.”
Line 205- 207, “The main goal of the resection of spinal tumors is to decompress neural structures to prevent and relieve neurological symptoms, reduce pain, and to achieve better functionality. Second goal is to achieve GTR whenever possible.” I don't understand what this means. GTR is the primary purpose of tumor resection.
What is the author's definition of "minimally invasive facetectomy and percutaneous instrumentation?" What makes the authors minimally invasive?" Is it time for surgery? and/or Is it the size of the skin incision?"
Line 225- 229, “However, spinal instrumentation with standard titanium implants may cause significant artifacts on MRI, which makes interpretation very difficult. To prevent this, we utilized CFR-PEEK hardware. This minimally metallic instrumentation (MMI) causes less artifacts, thus allowing for more adequate detection of potential tumor regrowth or other perioperative morbidities such as hematoma or infection [8]. “I am interested in this instrumentation, and please show the postoperative X-ray and MRI images.
Author Response
We thank the reviewer for their comments:
In the following, we answer the questions point by point:
Comments and Suggestions for Authors
Comments to the Author
I thank you for the opportunity to review your paper.
The authors evaluated the facet-sparing and facetectomy approaches for the treatment of lumbar spine dumbbell tumors. Nineteen patients were included. Nine patients were operated on using a facet-sparing technique. Ten patients underwent facetectomy and fusion. The relative risk (RR) for incomplete resection in the facet-sparing cohort was 18.7 (95% CI 1.23 - 284.047; p=0.035). The results showed that minimal invasive resection of lumbar Eden type 2 and 3 dumbbell tumors, including facetectomy in combination with instrumentation, appears safe and superior to the facet-sparing approach in local tumor control.
This is a topic of interest to spine surgeons. However, this paper has several issues to be resolved, such as sample size and indications in the surgical procedures, as the author mentions in the limitations. In order to turn this data into a valuable paper, it is necessary to increase the number of cases and decide on indications for surgical procedures. For example, what tumor size should a facetectomy approaches be performed for GTR?
Moreover, please present a case, including preoperative and postoperative X-ray and MRI images.
We enriched Figures 3-5 with images from representative cases of each cohort.
Line 72-73; “Preoperative tumor volumetric analysis was performed using Brainlab elements® (Brainlab AG, Munich, Germany) and expressed in milliliters (ml).” Please be specific, and it is easy to understand with a diagram. I think the tumor size may have affected the surgical technique.
The brain lab software allows for semiautomatic volumetric measurements of three-dimensional structures from standard MRI-imaging. This was clarified in the methods section. Concerning tumor size, please see above for answers to reviewer #1.
2.4 Surgical techniques; -There are no indications for the two procedures. How did you make each surgical decision?
The authors describe this in the results as follows. “Resection was performed according to surgeon's preference in line 139” This seems like a scientific problem.
The question of whether to perform a facetectomy (and fusion) is matter of debate. This work tries to elaborate on this comparing both two groups that formed by individual surgeon’s preference. Some attending surgeons on the team generally prefer a non-facetectomy technique. After recognizing the significant rate of re-surgery in the non-facetectomy cohort, a clinic internal consensus was made to address all Eden 3 and 4 types via facetectomy (and fusion). Indeed, our cohort does not constitute from a decision-making algorithm or randomization process. This is a major limitation, now acknowledged in the last paragrapgh of the discussion section.
Line 131-132, “From 2014 to 2021 two hundred and thirty (N=230) patients were treated surgically at our department for intraspinal extramedullary tumors. Of these, 82 were in the lumbar spine.” We don't need this data, and it should be removed.
This information is given to depict patient selection for this study, which is in accordance with recommendations provided by the STOBE reporting guideline. However, we abbreviated to “From 2014 to 2021 82 intra-spinal and extra-medullary lumbar tumors were treated surgically at our department. (…)”
Line 147- 150, “Gross total resection (GTR) of the tumor was achieved in all cases within the facetectomy cohort (N=10, 100%) in comparison to only one case in the facet-sparing cohort (N=1, 148 11,1%).” This result inevitably has a significant impact on the recurrence rate.
In the first place, there is a problem in comparing surgical results between two groups in which complete resection has not been performed.
The overall goal of this study was to investigate whether facetectomy influences extent of resection. GTR was achieved only in part in the non-facetectomy cohort. Because interpretation of postoperative images after resection of dumbbell tumors (usually scar tissue within the neuro-foramen enhances on contrast MRI) we decided also to see whether facetectomy harbors a higher rate of progressive disease and required second surgery. Still, we agree that extent of resection is a major confounder of progression measures. This was again highlighted in the last paragraph of the discussion section.
Line 203-204, “To the best of our knowledge, our study is the first comparative analysis on both 203 surgical approaches.”
We changed the sentence to: To the best of our knowledge, our study is the first one comparing both surgical approaches
Line 205- 207, “The main goal of the resection of spinal tumors is to decompress neural structures to prevent and relieve neurological symptoms, reduce pain, and to achieve better functionality. Second goal is to achieve GTR whenever possible.” I don't understand what this means. GTR is the primary purpose of tumor resection.
Sentence was deleted.
What is the author's definition of "minimally invasive facetectomy and percutaneous instrumentation?" What makes the authors minimally invasive?" Is it time for surgery? and/or Is it the size of the skin incision?"
The definition of minimally invasive spine surgery is indeed heterogeous. We decide to choose this term for the method mentioned in this study, because it is very similar to the more popular minimally invasive TLIF and to differentiate it from other more “open“ methods requiring wideranging bilateral disruption of the paraspinal muscles/ligaments and extensive laminectomy.
Line 225- 229, “However, spinal instrumentation with standard titanium implants may cause significant artifacts on MRI, which makes interpretation very difficult. To prevent this, we utilized CFR-PEEK hardware. This minimally metallic instrumentation (MMI) causes less artifacts, thus allowing for more adequate detection of potential tumor regrowth or other perioperative morbidities such as hematoma or infection [8]. “I am interested in this instrumentation, and please show the postoperative X-ray and MRI images.
We added exemplary images to the manuscript, please see above.
Round 2
Reviewer 2 Report
Dear Authors
Unfortunately, the author's comment does not adequately answer my comment. Moreover, I could not find any further figures or tables.
Reject
Author Response
Dear reviewer,
we thank you again for you time reviewing the paper.
maybe you could mention which comments are not adequately answered? Which methods
not describes adequately?
figures we’re added to the manuscript file
(word), and we’re uploaded to the sever.
Moreover, the file includes track changes.
However, I see the attached manuscript
is wrong, it’s submitted by other authors